# Anti-Inflammatory Effects of Low-Dose Glucocorticoids Compensate for Their Detrimental Effects on Bone Mineral Density in Patients with Rheumatoid Arthritis

**DOI:** 10.3390/jcm10132944

**Published:** 2021-06-30

**Authors:** Ji-Won Kim, Ju-Yang Jung, Hyoun-Ah Kim, Chang-Hee Suh

**Affiliations:** 1Department of Rheumatology, Ajou University School of Medicine, Suwon 16499, Korea; jwk722@naver.com (J.-W.K.); serinne20@hanmail.net (J.-Y.J.); nakhada@naver.com (H.-A.K.); 2Department of Molecular Science and Technology, Ajou University, Suwon 16499, Korea

**Keywords:** rheumatoid arthritis, glucocorticoids, osteoporosis, disease activity

## Abstract

Objectives: This study aimed to provide reliable information on the impact of low-dose glucocorticoids (GCs) on the bone mineral density (BMD) of patients with rheumatoid arthritis (RA). Methods: This retrospective study enrolled 933 patients with RA who continued the consumption of GCs (GC group) and 100 patients who had discontinued consumption for >1 year (no-GC group). The BMD values were measured at baseline and follow-up, and the annual rate of change in BMD between the groups was compared using dual-energy X-ray absorptiometry. We used multiple linear regression analysis to identify the factors associated with changes in BMD. Results: The demographic characteristics and use of medical treatments affecting bone metabolism were similar between the two groups. Furthermore, there were no significant differences in the annual rate of changes in BMD and incidence of newly developed osteoporosis and incidental fractures between the two groups. Multiple linear regression analysis revealed that the disease activity score for 28 joints with erythrocyte sedimentation rate was the only factor affecting the annual rate of changes in BMD, and it was inversely proportional to changes in BMD. Conclusion: The benefits of GC therapy in attenuating inflammation compensate for the risk of osteoporosis if adequate measures to prevent bone loss are implemented in patients with RA.

## 1. Introduction

Rheumatoid arthritis (RA) is a systemic autoimmune disease characterized by chronic, symmetrical, and progressive inflammatory polyarthritis [1]. Under pathological conditions of RA, the balance between bone resorption and formation is disrupted by the expression of pro-inflammatory cytokines that promote osteoclast differentiation and suppress the osteogenic activity of the osteoblasts [2]. Moreover, uncontrolled inflammation is transformed into hyperplastic invasive tissue, which destroys the cartilage and bone, leading to local and generalized bone loss [3]. This mechanism causes osteoporosis, a major complication of RA, which is a risk factor for fractures, impairs functional ability, deteriorates the quality of life, and further contributes to increased mortality [4].

Over the past few decades, glucocorticoids (GCs) have continued to play an important role in the treatment of various inflammatory diseases. GCs are widely recognized for their role as combination drugs in RA treatment and are considered highly effective in reducing the signs and symptoms of the disease and maintaining low disease activity [5]. However, GCs significantly contribute to the development of osteoporosis in patients with RA by decreasing the number of osteoblasts and promoting apoptosis of osteoblasts and osteocytes [6]. In most previous studies, GC-treated patients with RA had lower bone density and higher incidence rates of osteoporosis and fracture than the general population or patients with RA not treated with GCs [7]. However, high disease activity is another important risk factor that increases the incidence of osteoporosis, similar to GC therapy. Hence, it is often necessary to maintain low-dose GC therapy along with disease-modifying anti-rheumatic drugs (DMARDs) to achieve high and persistent remission rates [8]. 

Nevertheless, the prevailing opinion is that it is beneficial to discontinue GCs as soon as possible rather than maintain the remission state by administering low-dose GCs [9,10]. To provide reliable information on the effect of low-dose GCs on bone mineral density (BMD), we investigated the changes in BMD and disease activity, new-onset osteoporosis rates, and incidence of fractures during the maintenance of low-dose GC therapy in patients with RA. Furthermore, we analyzed other risk factors that could accelerate the reduction in BMD. 

## 2. Materials and Methods

### 2.1. Study Design and Population

In this retrospective study, we identified 1480 patients with a record of low-dose GC use (<7.5 mg/day of prednisone or equivalent) among patients with RA from the Ajou University Hospital medical records database [11]. We excluded 346 patients who did not meet the inclusion criteria or did not have sufficient data due to irregular visits. We reviewed data on 1234 eligible patients who visited Ajou University Hospital between January 1999 and June 2020. Finally, we included 933 patients, excluding those diagnosed with other rheumatic diseases or those receiving GCs for other causes. Figure 1 shows the flowchart of this study. The study group included 833 patients aged > 18 years who fulfilled the 1987 American College of Rheumatology (ACR) or 2010 ACR/European League Against Rheumatism classification criteria for RA diagnosis and continued to take GCs during the study period. The control group included patients diagnosed with RA who had not taken GCs for >1 year prior to the baseline BMD test. This study was approved by the Institutional Review Board of Ajou University Hospital (AJIRB-MED-MDB-21-109). 

### 2.2. Clinical Assessments

All participants underwent structured interviews, physical examinations, laboratory tests, medical record reviews, and radiologic tests. Demographic data included information on age, sex, body mass index (BMI), menopausal status, alcohol consumption, smoking habits, and concomitant diseases. Data on laboratory findings such as those for rheumatoid factor (RF), anti-citrullinated protein antibody, erythrocyte sedimentation rate (ESR), C-reactive protein (CRP), and biochemistry parameters of the bone were collected. The medical record reviews helped to identify the duration of the disease, clinical characteristics (tender/swollen joint count and patient global assessment), and medications during follow-up, and the disease activity score of 28 joints (DAS28). The details of the assessment and treatment of the disease were documented by the rheumatologists. All radiographs were assessed by a radiologist, and bone erosion was defined as the presence of erosion in one or more of the proximal interphalangeal joints, metacarpophalangeal joints, wrists, and metatarsophalangeal joints on radiographs of the hand and foot. The imaging file provided to the radiologists did not include clinical information; therefore, the reading was performed in a clinically blind state.

### 2.3. BMD and Fracture Measurements

The BMD of the lumbar spine (L1–L4) and proximal femur (femoral neck and total hip) was measured at study enrollment and follow-up by dual energy X-ray absorptiometry (GE Lunar, Madison, WI, USA). All the scanners were operated according to the manufacturer’s standard scanning software and positioning protocols. The absolute value of the BMD results is presented in grams per square centimeter (g/cm^2^) and additionally expressed as the T-score (compared to a normal young adult reference population) or Z-score (age-matched controls) according to the World Health Organization (WHO) criteria. According to the WHO criteria, BMD is classified as normal, osteopenia, or osteoporosis based on a T-score (or Z-score) of −1 or above, between −1 and −2.5 (or −2), and below −2.5 (or −2), respectively [12]. BMD loss per year and the annual rate of change in BMD were calculated as absolute values.

A prior history of fractures (vertebral and non-vertebral) was assessed by radiologic evaluation and using patient self-reporting questionnaires. If the plain radiograph appeared normal at baseline but showed a fracture at follow-up, then it was considered an incidental fracture. Patients with fractures not related to osteoporosis or pathological fractures due to malignancy were excluded from the analysis. 

### 2.4. Statistical Analysis

Data are presented as mean ± standard deviation. The Wilcoxon rank test, the chi-square test, or Fisher’s exact test was used to determine statistically significant differences in the variables between the GC and no-GC groups. Continuous variables were analyzed using the Wilcoxon rank test, and categorical variables were analyzed using the chi-square test or Fisher’s exact test. Multiple linear regression analysis was used to investigate the factors affecting the annual rate of change in BMD. All statistical analyses were performed using R software, version 3.6.1 (R studio, Boston, MA, USA). For all analyses, *p* < 0.05 was considered statistically significant.

## 3. Results

### 3.1. Patient Characteristics

The demographic and clinical characteristics of the patients at baseline are summarized in Table 1. There were 833 (89%) patients who consumed low-dose GCs (GC group) and 100 (11%) patients who had discontinued GCs for >1 year before the baseline BMD test (no-GC group). There was no significant difference between the two groups in terms of age, sex, menopausal status, BMI, smoking habits, alcohol consumption, or disease duration. However, compared to the no-GC group, the GC group had higher RF positivity, more pronounced elevation of DAS28-ESR, and the presence of bone erosion. The baseline bone biochemistry results did not deviate from the normal range and were similar between the two groups. There was a significant difference in the mean cumulative GC dose before BMD measurements between the groups, and the average doses were 7.68 ± 6.58 g and 5.99 ± 5.29 g of prednisone or an equivalent in the GC group and no-GC group, respectively. The mean daily GC dose before BMD measurements did not differ statistically between the groups (4.11 ± 2.6 mg/day vs. 3.61 ± 4.9 mg/day of prednisone or an equivalent; *p* = 0.716), and the mean GC usage during the follow-up period in the GC group was 2.23 mg/day of prednisone or an equivalent. Methotrexate (MTX) and biological agents were used marginally more frequently in the GC group than in the no-GC group. The use of DMARDs other than MTX and non-steroidal anti-inflammatory drugs was similar between the groups.

Table 2 presents the results of BMD measurements at baseline and osteoporosis treatments. The measurement results of the subregions are presented in grams per square centimeter (g/cm^2^) and as the T- or Z-score. The BMD of the lumbar spine was comparable between the groups, but that of the femoral neck and total hip was significantly lower in the GC group than in the no-GC group. There was no significant difference in the number of patients with osteoporosis according to the WHO criteria between the groups (407 (48.9%) vs. 39 (39%); *p* = 0.061). The proportion of patients treated with medications affecting bone metabolism, including vitamin D, calcium, and proton pump inhibitors, was similar in both groups. Bisphosphonates were the most commonly taken drugs for the treatment for osteoporosis in both groups, followed by selective estrogen receptor modulators, denosumab, and teriparatide.

### 3.2. Effects of Corticosteroid Exposure on BMD and Osteoporotic Fractures

The absolute differences between the baseline and follow-up BMD values and the annual rate of change in BMD at all measurement sites are presented in Table 3. In both groups, the BMD values of the lumbar spine increased, whereas those of the femoral neck and total hip decreased marginally, but without statistically significant differences between the groups. In the GC and no-GC groups, the annual rates of change in the BMD of the lumbar spine increased by 1.7% and 1.3%, respectively, whereas those of the femoral neck decreased by 0.33% and 0.53%, respectively, and those of the total hip decreased by 0.38% and 0.32%, respectively. However, no statistical differences were observed in the annual rate of change in BMD between the groups. During the follow-up period, the frequency of new osteoporosis was not significantly different between the GC group and the no-GC group (36 patients (4.3%) vs. 6 patients (6%); *p* = 0.441). A comparison of the fractures occurring prior to the baseline BMD test and the incidental fractures occurring during the study is shown in Figure 2. There were 43 (5.2%) new fractures in the GC group and 3 (3%) new fractures in the no-GC group, with no significant difference between them. Among them, vertebral fractures were noted in 37 patients in the GC group and 2 patients in the no-GC group, accounting for 4.4% and 2% of the total patient counts, respectively.

### 3.3. Factors Affecting Annual Changes in BMD

Various factors influencing the annual change in the BMD of the lumbar spine, femoral neck, and total hip were analyzed using multiple linear regression analysis, as presented in Table 4. Factors known to affect changes in BMD (such as age, menopause, BMI, and smoking) and DAS28-ESR or DAS28-CRP differences between baseline and follow-up were included in the variables. The comparison of the DAS28-ESR at baseline and follow-up showed that there were significant differences between the two groups. In the GC group, the DAS28-ESR decreased at follow-up compared to that at baseline, whereas in the no-GC group, it increased marginally at follow-up (−0.27 ± 1 vs. 0.83 ± 0.71; *p* < 0.001, data not shown). The mean change in the DAS28-CRP level was not significantly different between the groups. Univariate linear regression analyses revealed that age, menopause, BMI, baseline DAS28-ESR, ΔDAS28-ESR, cumulative GC dose, and dietary vitamin D intake were significantly associated with annual changes in BMD. Multiple linear regression analyses using stepwise variable selection showed that baseline DAS28-ESR and ΔDAS28-ESR were related to annual changes in the BMD of the lumbar spine. In the femoral neck and total hip, only ΔDAS28-ESR was an influential variable affecting annual changes in BMD.

### 3.4. Annual BMD Changes Depending on the GC Dose

Figure 3 shows the changes in BMD according to the mean daily GC dose in the GC group. Significant reductions in the BMD values of the femoral neck and total hip were observed in patients taking a dose of at least 2.5 mg/day compared to those taking a dose of <2.5 mg/day. In the ≥2.5 mg and <2.5 mg GC groups, the annual rates of change in the BMD of the femoral neck were −0.84 ± 3.63 and 0.01 ± 4.44, respectively (*p* = 0.004), and those of the total hip were −0.67 ± 3.44 and −0.19 ± 3.44, respectively (*p* = 0.048). There was no difference in the annual rate of change in the BMD of the lumbar spine according to the GC dose.

## 4. Discussion

It is widely known that GCs can cause bone loss, reduce bone strength, and eventually lead to osteoporotic fractures; however, they exhibit protective roles due to their anti-inflammatory actions against underlying diseases [13]. Therefore, the debate regarding the balance between the benefits and risks of GC therapy is ongoing, with no consensus regarding either opinion since their introduction in the 1950s. As we already know, GCs are widely used in chronic diseases because of their rapid anti-inflammatory effects [14,15,16]. Especially in RA, the treatment guidelines suggest the use of GCs for the shortest period possible as a bridging therapy until conventional DMARDs show efficacy (<3 months) [17]. In actual clinical practice, the reduction or withdrawal of GCs is failing for various reasons; therefore, it is necessary to identify the deleterious effects of even low-dose GCs. This study analyzed the changes in BMD and the incidence of osteoporotic fractures among several adverse effects. To determine the effect of low-dose GCs on BMD, we compared patients undergoing low-dose GC therapy with those who discontinued GC therapy, rather than dividing them into never-/ever-exposed groups, because most patients receive GCs at the beginning of RA treatment. Although the no-GC group comprised past users rather than non-users, the risk from GCs would have had a minimal impact on the results of the study since the risk of osteoporosis and fractures decreases after cessation of GC therapy, and most participants were enrolled after a year of discontinuation of GCs [18].

The baseline disease activity, cumulative GC doses, and radiologic changes in the GC group increased compared to those in the no-GC group, which could be the reason for the inability to discontinue GC. The clinical difference between the two groups did not affect the BMD of the lumbar spine; however, there was a significant difference in the baseline BMD values of the femoral neck and total hip between the groups. Despite the high cumulative GC doses in the GC group, the result that the BMD of only some subregions decreased was inconsistent with that reported in a previous study [19]. In a previous meta-analysis, there was no disagreement on the conclusion that the BMD of all areas decreased with GC therapy; however, recently, other factors such as anti-bone-resorption agents and disease activity have been reported to outperform the bone-reduction effect of GCs [20,21]. Several studies on patients with RA treated with GCs have shown that the use of anti-bone-resorption agents increases the BMD and markers of bone turnover of the lumbar spine [22,23]. Another study showed that the markers of bone metabolism have different influencing abilities depending on the skeletal sites; thus, BMD does not decrease uniformly in all areas [24]. Although the effect on cumulative GC doses cannot be excluded, it may be necessary to closely analyze the relationship between the pathophysiology of RA and bone metabolism, as there have been studies in the literature to indicate that patients with RA have lesser changes in lumbar BMD compared to other subregions [25]. 

In both groups, the annual rate of change in the BMD of the lumbar spine increased, while that of the femoral neck and total hip decreased. In many older adults, degenerative change in the lumbar spine increases its BMD falsely, rather than causing an actual increase [26]. Notably, unlike in previous studies, low-dose GCs did not significantly affect the annual rate of change in BMD in our study. There was no increase in the incidence of new-onset osteoporosis or osteoporotic fractures during the follow-up period in the GC group. This is in contrast to the hypothesis that the use of GCs in patients with RA increases the risk of osteoporosis and osteoporotic fractures [27,28]. Similarly to our study, other studies have advocated the use of low-dose GCs in patients with RA based on bone loss [29,30,31,32,33]. This emphasizes that our findings are reliable, because previous studies included patients with early RA or had small samples or a short follow-up period [34]. Considering this, the long-term follow-up observation in this study is a significant advantage.

From the above results alone, it cannot be concluded that low-dose GCs have no effect on BMD reduction. Given the differences in the baseline characteristics between the two groups, factors other than GCs may have affected the annual rate of change in BMD. Several independent factors associated with the decreasing annual rate of change in BMD were identified using univariate statistical analysis. The analysis showed that ΔDAS28-ESR and cumulative GC dose were associated with the annual rates of change in the BMD of the lumbar spine, femoral neck, and total hip. In addition, age, menopausal status, BMI, baseline DAS28-ESR, and use of vitamin D supplements were found to be risk factors for some of the subregions. These findings are similar to those of previous studies [35,36,37,38,39]. In the multivariate approach calibrating the interrelated variables, only baseline DAS28-ESR and ΔDAS28-ESR contributed to BMD reduction. Thus, among the potential factors, achieving remission status by controlling the disease activity is the most critical factor in preventing bone loss. This result is consistent with the findings that tighter control of disease activity is the greatest contributor in improving the outcome of RA with the discovery of various biological agents for the treatment of RA over the past 30 years [40]. Data from the past decade have shown a potential positive effect on BMD in patients with RA whose disease activity is well-controlled by the use of biological agents, regardless of GC therapy [41]. In a recent study, RA patients with persistent remission status displayed similar changes in BMD even when compared with normal controls [42,43]. Furthermore, several studies conducted on other chronic inflammatory diseases, such as systemic lupus erythematosus and asthma, have also shown that proper management of the disease is clinically important to preventing BMD reduction, rather than discontinuing the use of GCs [44,45].

Since there were too many factors affecting the disease activity of RA and bone metabolism, we performed further analysis to minimize bias in the interpretation of the results. To compensate for the differences in baseline characteristics of the two groups, we additionally conducted propensity score matching (PSM) with factors that directly affect osteoporosis and fracture risk, including age, sex, seropositivity, and DMARDs. Comparison by PSM also showed no differences in the annual rate of change in BMD between the GC group and the no-GC group. Furthermore, the result that ΔDAS28-ESR was an independent factor influencing the annual rate of change in the BMD remained unchanged after PSM (Appendix A). Despite the effects of confounding factors, the results of this study were similar to those after PSM. Collectively, various factors affecting bone metabolism in patients with RA did not ultimately change the effect on GC-induced BMD reduction and fracture risk.

As mentioned above, it is essential to maintain a well-controlled status in chronic diseases, which requires continuous management after diagnosis; however, this does not mean that the use of GCs is unconditionally justified. In addition to the incidence of osteoporosis and osteoporotic fractures, GCs are associated with the occurrence of other complications such as hyperglycemia, hypertension, glaucoma, and mood changes; therefore, personalized treatment strategies should be developed. Just as the effectiveness of RA treatments has been dramatically improved by introducing paradigm shifts in applying treat-to-target strategies as a personalized approach, introducing a personalized approach can lead to various benefits in the use of GCs as well [46]. Thus, our study further analyzed whether there was a dose-dependent change in BMD. In line with recommended guidelines to prevent further fractures in all patients with previous osteoporosis fractures taking glucocorticoids, a very-low-GC group was classified based on 2.5 mg prednisone or an equivalent [47]. We found that even among patients consuming doses of ≤7.5 mg, which is generally considered a low dose, there was a significantly lesser decrease in the annual rate of change in BMD in the <2.5 mg GC group than in the ≥2.5 mg GC group. This means that a very low dose of GC (2.5 mg) does not significantly decrease BMD, but it has a favorable effect on maintaining the remission status of RA. A previous study reported that the incidence of vertebral and non-vertebral fractures in patients with RA may be higher, even in those treated with an oral GC dose of 2.5 mg, than that in healthy controls, and this analysis was criticized as having a limitation of not considering the disease activity of RA [18]. In addition, a study conducted on RA patients who achieved clinical remission with the first biologic DMARD reported that the group who stopped GC had a longer survival time on biologic DMARD [48]. However, since this study included relatively young patients using biologic DMARD, these results cannot be broadly interpreted as indicating a requirement for GC to be discontinued to reach clinical remission in all RA patients.

Our study is novel because it is the first study to report that very-low-dose GCs (<2.5 mg/day) are relatively safe in terms of controlling the disease activity and preventing osteoporosis and fractures. It is best to maintain clinical remission while reducing the use of GCs as much as possible through various combinations of medications; however, there is no need to discontinue GCs forcefully due to concerns of its adverse effects if the inflammation is not fully controlled. Furthermore, maintaining very-low-dose GC therapy requires meeting other conditions to prevent BMD reduction, such as calcium and vitamin D supplementation and appropriate pharmacological treatment of osteoporosis [49]. Finally, this study concludes that the benefits of GC therapy in attenuating inflammation compensate for the risk of osteoporosis when sufficient preventive measures are taken to prevent bone loss in patients with RA.

The limitations of our study include its retrospective single-center design. Due to these limitations, there were differences in the baseline characteristics and some DMARDs of the two groups. Factors affecting osteoporosis and fracture risk due to direct effects on bones, such as anti-TNF agents, bisphosphonate, and autoantibody positivity, were
also different. Autoantibody positivity is known to be a negative prognostic factor for bone erosions and osteoporosis, as well as RA; therefore, patients with seropositive RA would have needed to maintain GC to achieve low disease activity or remission status. To overcome this limitation, we additionally conducted propensity score matching (PSM) for age, sex, seropositivity, and some DMARDs, which showed no differences in the annual rate of change in BMD between the GC group and the no-GC group. Nevertheless, the inability to match all demographic and clinical characteristics of patients with RA enrolled in this study remained a limitation. Another limitation is that the study did not include all patients with RA taking GCs, but only those who underwent BMD testing. Therefore, the proportion of postmenopausal women increased compared to the known prevalence of RA by sex and age, which inevitably served as a selection bias. Moreover, osteoporosis-preventive treatment was not administrated equally to all patients, so the dosage of vitamin D or calcium varied among patients. In addition, functional disability, which affects the occurrence of fractures, was not evaluated. This study covered a vast period of 20 years, during which there have been many advances in medications; thus, the frequency of use of the most recent medications, such as teriparatide and denosumab for osteoporosis treatment, and JAK inhibitors for RA treatment, was relatively low.

## 5. Conclusions

Our study suggest that the net effect of low-dose GCs compensates for its detrimental effects on BMD in patients with RA. Tight control of disease activity, even with GC therapy, may reduce the long-term effects of bone inflammation and halt progressive bone loss. In conclusion, it is recommended that rheumatologists optimize the use of GCs to minimize the associated adverse effects, including osteoporosis and osteoporotic fractures, while maintaining clinical remission or improving anti-inflammatory efficacy through GC therapy. 

## Figures and Tables

**Figure 1 jcm-10-02944-f001:**
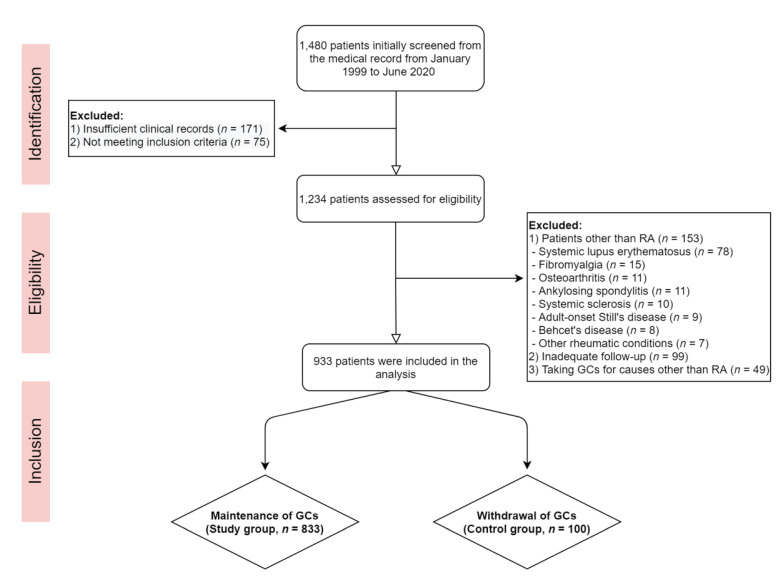
Flowchart of this retrospective study. RA: Rheumatoid arthritis, GCs: Glucocorticoids.

**Figure 2 jcm-10-02944-f002:**
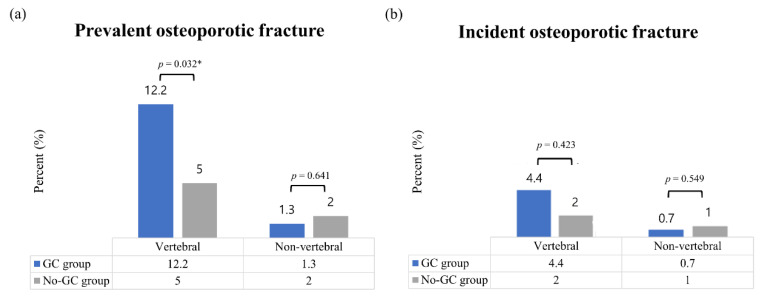
Comparison of vertebral and non-vertebral osteoporotic fractures. (**a**) Prevalent vertebral and non-vertebral fractures at baseline; (**b**) Incidental vertebral and non-vertebral fractures after baseline. GC: Glucocorticoid. * *p* < 0.05.

**Figure 3 jcm-10-02944-f003:**
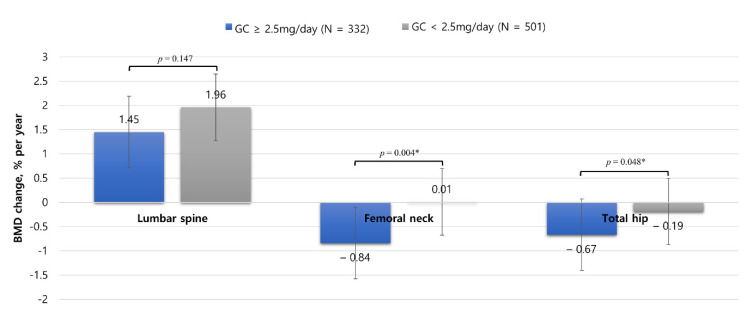
Annual rates of change in the bone mineral density of the lumbar spine, femoral neck, and total hip according to the mean daily glucocorticoid dose. Error bars represent the standard error of the mean. GC: Glucocorticoid, BMD: bone mineral density. * *p* < 0.05.

**Table 1 jcm-10-02944-t001:** Baseline demographic and clinical characteristics of patients with RA.

Variable	GC (N = 833)	No-GC (N = 100)	*p* Value
Age, mean (years)	63.7 ± 9.8	61.8 ± 9	0.062
Sex			0.15
Women, N. (%)	766 (92)	96 (96)	
Menopause, N. (%)	694 (90.7)	85 (88.5)	0.493
BMI, mean	23.3 ± 3.72	23.8 ± 3.33	0.208
Smoking, N. (%)	48 (5.8)	9 (9)	0.203
Alcohol, N. (%)	91 (10.9)	11 (11)	0.985
RF positivity, N. (%)	563 (67.3)	51 (51)	0.001 *
Anti-CCP Ab positivity, N. (%)	300 (36)	38 (38)	0.181
Disease duration (months)	110.7 ± 77.6	100.2 ± 58.8	0.22
DAS28-ESR	3.02 ± 1.5	2.56 ± 0.9	0.012 *
DAS28-CRP	2.19 ± 0.98	1.86 ± 0.65	0.062
Bone erosion, N. (%)	329 (39.4)	21 (21)	<0.001 *
Bone biochemistry			
Serum 25 (OH) D (ng/ml) (normal range 30.0‒100.0)	27.2 ± 20.7	29.3 ± 11.3	0.415
Serum calcium, (mg/dL) (normal range 8.6‒10.2)	9.52 ± 4.28	9.43 ± 0.32	0.834
Serum phosphate, (mg/dL) (normal range 2.5‒4.5)	3.52 ± 0.56	3.64 ± 0.46	0.054
Serum creatinine, (mg/dL) (normal range 0.5‒0.9)	0.78 ± 0.39	0.73 ± 0.13	0.085
Serum alkaline phosphatase, (U/L) (normal range 35‒104)	66.6 ± 22.8	66.4 ± 19.8	0.93
Medication			
Glucocorticoids			
Cumulative dose before BMD measurements (g)	7.68 ± 6.58	5.99 ± 5.29	0.004 *
Mean dose before BMD measurements (mg/day)	4.11 ± 2.6	3.61 ± 4.9	0.716
Mean dose between BMD measurements (mg/day)	2.23 ± 1.69	NA	NA
NSAIDs, N. (%)	560 (67.2)	61 (61)	0.219
Methotrexate, N. (%)	549 (66)	39 (39)	<0.001 *
Hydroxychloroquine, N. (%)	500 (60)	70 (70)	0.053
Sulfasalazine, N. (%)	53 (6.4)	9 (9)	0.317
Tacrolimus/cyclosporin, N. (%)	174 (20.9)	13 (13)	0.063
Leflunomide, N. (%)	144 (17.2)	11 (11)	0.119
Anti-TNF agents, N. (%)	38 (4.6)	1 (1)	0.048 *
Adalimumab, N. (%)	10 (1.2)	0 (0)	0.612
Etanercept, N. (%)	9 (1.1)	1 (1)	>0.999
Golimumab, N. (%)	8 (1)	0 (0)	>0.999
Infliximab, N. (%)	11 (1.3)	0 (0)	0.619
Abatacept, N. (%)	4 (0.5)	0 (0)	0.34
Tocilizumab, N. (%)	17 (2.0)	0 (0)	0.242
Rituximab, N. (%)	8 (1)	0 (0)	>0.999
JAK inhibitors, N. (%)	11 (1.3)	0 (0)	0.619

RA: rheumatoid arthritis, GC: glucocorticoid, BMI: body mass index, RF: rheumatoid factor, Anti-CCP Ab: anti-citrullinated protein antibody, DAS: disease activity score, ESR: erythrocyte sedimentation rate, CRP: C-reactive protein, 25(OH) D: 25-hydroxyvitamin D, BMD: bone mineral density, NSAIDs: non-steroidal anti-inflammatory drugs, TNF: tumor necrosis factor, JAK: Janus kinase, NA: not applicable. * *p* < 0.05.

**Table 2 jcm-10-02944-t002:** Baseline bone mineral density and osteoporosis treatments of patients with RA.

Variable	GC (N = 833)	No-GC (N = 100)	*p* Value
Baseline BMD (g/cm^2^)			
Lumbar spine	0.94 ± 0.16	0.95 ± 0.16	0.813
Femoral neck	0.75 ± 0.12	0.78 ± 0.11	0.022 *
Total hip	0.8 ± 0.14	0.85 ± 0.12	0.001 *
Baseline BMD, T-score			
Lumbar spine	−1.83 ± 1.29	−1.64 ± 1.4	0.55
Femoral neck	−1.55 ± 0.98	−1.33 ± 0.88	0.033 *
Total hip	−1.38 ± 1.13	−0.99 ± 1.01	0.002 *
WHO classification using BMD measurements			
Normal, N. (%)	107 (12.8)	13 (13)	0.965
Osteopenia, N. (%)	319 (38.3)	48 (48)	0.06
Osteoporosis, N. (%)	407 (48.9)	39 (39)	0.062
Interval between BMDs (months)	19.5 ± 12.1	19.9 ± 12	0.538
Calcium intake, N. (%)	317 (38.1)	47 (47)	0.103
Vitamin D intake, N. (%)	724 (86.9)	94 (94)	0.053
Proton pump inhibitor, N. (%)	170 (20.4)	16 (16)	0.354
Treatment for osteoporosis, N. (%)	411 (49.3)	40 (40)	0.077
Bisphosphonate, N. (%)	323 (78.6)	28 (70)	0.036 *
SERM, N. (%)	70 (17)	11 (27.5)	0.384
Denosumab, N. (%)	26 (6.3)	1 (3.6)	0.348
Teriparatide, N. (%)	2 (0.5)	0 (0)	>0.999

RA: rheumatoid arthritis, GC: glucocorticoid, BMD: bone densitometry, WHO: World Health Organization, SERM: selective estrogen receptor modulator. * *p* < 0.05.

**Table 3 jcm-10-02944-t003:** Annual change in BMD and new-onset osteoporosis rates between the GC users and the control group.

Variable	GC (N = 833)	No-GC (N = 100)	*p* Value
Annual mean value of the absolute difference between baseline and follow-up BMD (g/cm^2^)			
Lumbar spine	0.014 ± 0.043	0.01 ± 0.042	0.414
Femoral neck	−0.003 ± 0.028	−0.005 ± 0.02	0.959
Total hip	−0.003 ± 0.031	−0.003 ± 0.015	0.6
Annual (%) change in BMD (g/cm^2^)			
Lumbar spine	1.76 ± 4.9	1.26 ± 4.44	0.393
Femoral neck	−0.33 ± 4.16	−0.53 ± 2.62	0.6
Total hip	−0.38 ± 3.42	−0.32 ± 1.8	0.453
New-onset osteoporosis, N. (%)	36 (4.3)	6 (6)	0.441

BMD: bone densitometry, GC: glucocorticoid.

**Table 4 jcm-10-02944-t004:** Multiple linear regression analysis of risk factors associated with annualized BMD change.

Variable	Lumbar Spine		Femoral Neck		Total Hip	
	β	*p* Value	β	*p* Value	β	*p* Value
Univariate						
Age	0.041	0.013 *	−0.001	0.942	0	0.998
Menopause	1.560	0.006 *	−0.387	0.421	−0.003	0.994
BMI	−0.027	0.532	−0.033	0.362	−0.078	0.009 *
Smoking	1.01	0.136	−0.188	0.734	−0.088	0.846
Alcohol	0.001	0.999	−0.01	0.98	0.008	0.982
RF positivity	0.085	0.801	−0.001	0.998	−0.03	0.897
Anti-CCP Ab positivity	0.228	0.574	0.158	0.644	0.27	0.302
Disease duration	0.001	0.543	−0.001	0.583	0.001	0.474
Baseline DAS28-ESR	0.59	0.002 *	0.457	0.009 *	0.218	0.099
Baseline DAS28-CRP	0.379	0.109	0.192	0.369	0.048	0.768
ΔDAS28-ESR	−1.9	< 0.001 *	−0.936	<0.001 *	−1.02	<0.001 *
ΔDAS28-CRP	0.379	0.109	0.192	0.369	0.048	0.768
Bone erosion	0.538	0.103	−0.008	0.769	0.129	0.565
Cumulative CS dose	0.077	0.002 *	0.029	0.163	0.113	0.002 *
Calcium intake	−0.344	0.308	0.209	0.455	0.673	0.621
Vitamin D intake	1.61	0.001 *	0.964	0.016	0.052	0.04 *
PPI use	0.189	0.856	0.01	0.114	0.599	0.638
Multivariate						
Age	0.009	0.707	0.009	0.659	0.002	0.885
Menopause	−0.077	0.917	−0.131	0.839	−0.14	0.772
BMI	−0.031	0.585	−0.047	0.367	−0.018	0.637
Baseline DAS28-ESR	−0.593	0.006 *	−0.068	0.737	0.223	0.092
ΔDAS28-ESR	−2.08	<0.001 *	−0.908	<0.001 *	−0.964	<0.001 *
Cumulative GC dose	−0.003	0.914	0.051	0.072	−0.003	0.906
Vitamin D intake	−0.119	0.842	0.619	0.26	0.309	0.445

BMD: bone mineral density, BMI: body mass index, RF: rheumatoid factor, Anti-CCP Ab: anti-citrullinated protein antibody, ESR: erythrocyte sedimentation rate, CRP: C-reactive protein, DAS: disease activity score, GC: glucocorticoid, PPI: proton pump inhibitor, NA: not applicable. ΔDAS28-ESR is DAS28-ESR at the follow-up point minus baseline DAS28-ESR. ΔDAS28-CRP is DAS28-CRP at the follow-up point minus baseline DAS28-CRP. * *p* < 0.05.

## Data Availability

The datasets generated for this study are not available due to the data protection law.

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
