# Peer review of "Anti-Inflammatory Effects of Low-Dose Glucocorticoids Compensate for Their Detrimental Effects on Bone Mineral Density in Patients with Rheumatoid Arthritis"

_jcm, 2021, doi:10.3390/jcm10132944_

Round 1

Reviewer 1 Report

The manuscript is written well in its present form

Author Response

Thank you for your positive comment. 

Reviewer 2 Report

Manuiscript's version improved after revision.

Please comment in Discussion the effect to reduce or even stopping GC (according to EULAR reccomendation for RA management and article from Fornaro et al. Eur J Clin Invest. 2021 Feb;51(2):e13363 stopping GC as predictor of long term clinical remission).

Correct in Table 4 "CS" with "GC"

In figure 2. use the same order axis unit both for "prevalent" and "Incident" osteoporotic fractures.

Author Response

Please comment in Discussion the effect to reduce or even stopping GC (according to EULAR reccomendation for RA management and article from Fornaro et al. Eur J Clin Invest. 2021 Feb;51(2):e13363 stopping GC as predictor of long term clinical remission).

Answer> Thank you for your comment. We added the following comment to the discussion referring to the literature you mentioned. The added parts were underlined and changed in color.

  • In addition, a study conducted on RA patients who achieved clinical remission with the first biologic DMARD reported that the group who stopped GC had a longer survival on biologic DMARD [48]. However, since this study included relatively young patients using biologic DMARD, these results cannot be broadly interpreted as requiring GC to be discontinued to reach clinical remission in all RA patients.

Correct in Table 4 "CS" with "GC"

Answer> Thank you for your comment. As your comment, we changed table 4 and underlined the changes.

In figure 2. use the same order axis unit both for "prevalent" and "Incident" osteoporotic fractures.

Answer> Thank you for the good comment. Figure 2 was reconstructed using the same order axis unit for prevalent and incident osteoporotic fractures.

Thank you for the constructive review. We hope that the revised manuscript now meets the journal’s standards for publication.

This manuscript is a resubmission of an earlier submission. The following is a list of the peer review reports and author responses from that submission.

Round 1

Reviewer 1 Report

Dear Authors,

first of all, I thank You for giving me the opportunity to read this Your manuscript, submitted for publication.

Here are my comments and suggestions.

MAJOR COMMENT

In Table 1, You inserted biological and synthethic DMARDS. However, DMARDS can have direct effects on bone. This could be a bias.  Please, clarify and discuss this point.

MINOR COMMENTS

a) In the synovial tissue there is no bone resorption and formation, as You wrote. Please remove this sentence.

b) Please, remove lines from 39 to 41 because not useful in the Introduction section.

c) You wrote "Nevertheless, the prevaling opinion is that it is beneficial...." Please, add some references. 

d) Study design and population: "We excluded 346 patients who did not met the inclusion criteria...". They must be specified here and not later. Do You agree ?

e) Discussion seemed repetitive in several points. Please, consider to re-write it in a more linear way. 

Author Response

Dear Reviewer,

We appreciate your review of our manuscript “Anti-inflammatory effects of low-dose glucocorticoids compensate for their detrimental effects on bone mineral density in patients with rheumatoid arthritis”. In response to your comments and those of the reviewers, we have made several changes to the text, as summarized below.

MAJOR COMMENT

In Table 1, You inserted biological and synthethic DMARDS. However, DMARDS can have direct effects on bone. This could be a bias.  Please, clarify and discuss this point.

Answer> Thank you for your comment. Biological and synthetic DMARDS cannot be excluded as a treatment for rheumatoid arthritis (RA). As you mentioned, biological and synthetic DMARDS have direct effects on bones, which can influence bone mineral density, however, when choosing medications for RA, various problems such as side effects and comorbidities are considered. That is why we thought it would be rather a bias to choose patients considering medications that affect bone mineral density. We recognized the limitation; therefore we reinforced this section as follows and marked with red-color.

  • The limitations of our study include its retrospective single-center design. Due to these limitations, there were differences in the baseline characteristics and some DMARDs of the two groups. Factors affecting osteoporosis and fracture risk due to direct effects on bones, such as anti-TNF agents, bisphosphonate and autoantibody positivity, were also different. Although there are many factors in the GC group that have a protective effect on osteoporosis, it cannot be determined that they acted in a favorable direction to reduce bone density as the rate of bone erosion at the baseline was high.

MINOR COMMENTS

  1. a) In the synovial tissue there is no bone resorption and formation, as You wrote. Please remove this sentence.

Answer> Thank you for your comment. We modified sentence as follows and marked with red-color.

  • Under pathological conditions of RA, the balance between bone resorption and formation is disrupted by the expression of pro-inflammatory cytokines that promote osteoclast differentiation and suppress the osteogenic activity of the osteoblasts.
  1. b) Please, remove lines from 39 to 41 because not useful in the Introduction section.

Answer> Thank you for your comment. We removed the lines from 39 to 41 you mentioned.

  1. c) You wrote "Nevertheless, the prevaling opinion is that it is beneficial...." Please, add some references. 

Answer> Thank you for your comment. We have added the following references.

  • van Everdingen AA, Jacobs JW, Siewertsz Van Reesema DR, et al. Low-dose glucocorticoids in early rheumatoid arthritis: Discordant effects on bone mineral density and fractures? Clin Exp Rheumatol. 2003;21(2):155-60.
  • Cheng T-T, Lai H-M, Yu S-F, et al. The impact of low-dose glucocorticoids on disease activity, bone mineral density, osteoporotic fractures, and 10-year probability of fractures in patients with rheumatoid arthritis. J Investig Med. 2018;66(6):1004-7.

  1. d) Study design and population: "We excluded 346 patients who did not met the inclusion criteria...". They must be specified here and not later. Do You agree ?

Answer> We totally agree with your comment. We added the details and revised it as follows and marked with red-color.

  • We excluded 346 patients who did not meet the inclusion criteria or did not have sufficient data due to irregular visit.

  1. e) Discussion seemed repetitive in several points. Please, consider to re-write it in a more linear way. 

Answer> Thank you for your valuable comment. We described the sentence more clearly. In addition, we have changed the overall content of the discussion to be refined and concise. Many contents were changed; therefore we could not write it here. The change part was marked with red color.

Thank you for the constructive review. We hope that the revised manuscript now meets the journal’s standards for publication.

Reviewer 2 Report

In this article the Authors evaluated the effect on bone mineral density of low-dose (< 7.5 mg/day) steroids compared with no steroid intake since 1 year in rheumatoid arthritis patients.

The Authors conclude that the benefits of low-dose steroid therapy in controlling disease compensate the risk of bone mineral density reduction and fractures in rheumatoid arthirtis patients if measures to prevent osteoporsis are carried out.

The study background, methods, statistical analysis, results and conclusions are correct.

Minor comments: in the Methods section (2.2 Clinical assessment) clarify whether radiologists and rheumatologists were aware if patients were taking (or not) steroid therapy.

Author Response

Dear Reviewer,

We appreciate your review of our manuscript “Anti-inflammatory effects of low-dose glucocorticoids compensate for their detrimental effects on bone mineral density in patients with rheumatoid arthritis”. In response to your comments and those of the reviewers, we have made several changes to the text, as summarized below.

Reviewer: 2
In this article the Authors evaluated the effect on bone mineral density of low-dose (< 7.5 mg/day) steroids compared with no steroid intake since 1 year in rheumatoid arthritis patients.

The Authors conclude that the benefits of low-dose steroid therapy in controlling disease compensate the risk of bone mineral density reduction and fractures in rheumatoid arthirtis patients if measures to prevent osteoporosis are carried out.

The study background, methods, statistical analysis, results and conclusions are correct.

Minor comments: in the Methods section (2.2 Clinical assessment) clarify whether radiologists and rheumatologists were aware if patients were taking (or not) steroid therapy.

Answer> Thank you for your comment. We have attached methods section about whether to recognized steroid use as follows and marked with red-color.

The imaging file provided to the radiologists did not include clinical information, therefore the reading was performed in a clinically blind state. 

Thank you for the constructive review. We hope that the revised manuscript now meets the journal’s standards for publication.

Reviewer 3 Report

Interesting research but some points need to be clarified.

RF positivity is known to be a negative prognostic factor for bone erosions and these to osteoporosis and fractures’ risk. GC patients presented significant higher prevalence of RF positivity, higher disease activity and above all more bone erosions already at baseline compared to non-GC group. Moreover, GC patients were more frequently treated with MTX, anti-TNF agents and bisphosphonate.  These drugs reduce bone resorption and have protective effects on osteoporosis.

All these features represent relevant selection bias, that influenced authors’ findings.

Finally, is not clear why the authors choose the 2.5 mg/day as cut-off for low dose GC definition.

Please clarify and comment.

Please, English should be revised by a native speaker.

Author Response

Dear Reviewer,

We appreciate your review of our manuscript “Anti-inflammatory effects of low-dose glucocorticoids compensate for their detrimental effects on bone mineral density in patients with rheumatoid arthritis”. In response to your comments and those of the reviewers, we have made several changes to the text, as summarized below.

Reviewer: 3

Interesting research but some points need to be clarified.

RF positivity is known to be a negative prognostic factor for bone erosions and these to osteoporosis and fractures’ risk. GC patients presented significant higher prevalence of RF positivity, higher disease activity and above all more bone erosions already at baseline compared to non-GC group. Moreover, GC patients were more frequently treated with MTX, anti-TNF agents and bisphosphonate.  These drugs reduce bone resorption and have protective effects on osteoporosis.

All these features represent relevant selection bias, that influenced authors’ findings.

Answer> Thank you for your comment. What you have pointed out is the inevitable limitations for retrospective study. Although we were aware of the limitations, we thought that it was rather a selection bias to conduct the study after matching all the characteristics, so we proceeded the study without selecting patients. One of the highlights in this study is that there was no change in annual change bone mineral density if disease activity was well-controlled despite the difference in baseline characteristics. For this reason, the limitations are stated in the text as follows and marked with red-color.

  • The limitations of our study include its retrospective single-center design. Due to these limitations, there were differences in the baseline characteristics and some DMARDs of the two groups.

Finally, is not clear why the authors choose the 2.5 mg/day as cut-off for low dose GC definition. Please clarify and comment.

Answer> Thank you for your comment. If GC use is included as a risk factor in FRAX (fracture risk assessment tool), a fracture risk occurred at a dose of GC 2.5-7.5mg/day. Based on these results, recommendation guidelines for the prevention of additional fractures in any patients with previous osteoporotic fractures taking glucocorticoids have been published based on GC 2.5mg (Buckley L et al., Arthritis Rheumatol, 2017). Referring to the previous content, we choose 2.5mg/day as cut-off for very low-dose GC definition. As you mentioned, the content related to the cut-off for very low-dose GC definition was not clear, therefore, we added the sentence as follows.

  • In line with the recommended guidelines to prevent further fractures in all patients with previous osteoporosis fractures taking glucocorticoids, the very low-dose GC group was classified based on GC 2.5mg.

\

Please, English should be revised by a native speaker.

Answer> Thank you for your comment. As your comment, we shorted the manuscript to be concise. Our manuscript was already checked by fluent English speaker. We used Editage (www.editage.co.kr) for English language editing.

Thank you for the constructive review. We hope that the revised manuscript now meets the journal’s standards for publication.

Round 2

Reviewer 1 Report

Dear Authors,

all my comments and suggestions were satisfactorily met in the revised version of Your manuscript. 

Best regards. 

Author Response

Thank you for your positive comment.